# A 54 µW CMOS Auto-Trimming Bandgap References (ATBGR) Achieving 90 dB PSRR for Artificial Intelligence of Things (AIoT) Chips

**DOI:** 10.3390/mi14091724

**Published:** 2023-09-01

**Authors:** Balamahesn Poongan, Jagadheswaran Rajendran, Selvakumar Mariappan, Arvind Singh Rawat, Narendra Kumar, Arokia Nathan, Binboga S. Yarman

**Affiliations:** 1Collaborative Microelectronic Design Excellence Center (CEDEC), Universiti Sains Malaysia, Bayan Lepas 11900, Malaysia; balamahesn.poongan@student.usm.my (B.P.); selva_kumar@usm.my (S.M.); 2School of Computing, DIT University, Dehradun 248009, Uttarakhand, India; arvindsrawat@ieee.org; 3Department of Electrical Engineering, Faculty of Engineering, University of Malaya, Kuala Lumpur 50603, Malaysia; narendra.k@um.edu.my; 4Darwin College, Cambridge University, Cambridge CB3 9EU, UK; an299@cam.ac.uk; 5Department of Electrical and Electronics Engineering, Istanbul University, 34320 Istanbul, Turkey; sbyarman@gmail.com

**Keywords:** bandgap voltage reference (BGR), power supply rejection ratio (PSRR), artificial intelligence and Internet of Things (AIoT), CMOS

## Abstract

An Auto-Trimming CMOS Bandgap References Circuit (ATBGR) with PSRR enhancement circuit for Artificial Intelligence of Things (AIoT) chips is presented in this paper. The ATBGR is designed with a first-order temperature compensation technique providing a stable reference voltage of 1.25 V in the ranges of input voltages from 1.65 V to 4.5 V. An auto-trimming circuit is integrated into a PTAT resistor of BGR to minimize the influences of the process variations. The four parallel resistor pairs with PMOS switches are connected in series with the PTAT resistor. The reference voltage, *V*_REF_, is compared to an external constant value, 1.25 V, through an operational amplifier, and the output of the de-multiplexer is used to configure the PMOS switches. High power supply rejection is achieved through a PSRR enhancement circuit constituting a cascaded PMOS common gate pair. The ATBGR circuit is fabricated in 180 nm CMOS technology, consuming an area of 0.03277 mm^2^. The auto-trimming method yields an average temperature coefficient of 9.99 ppm/°C with temperature ranges from −40 °C to 125 °C, and a power supply rejection ratio of −90 dB at 100 MHz is obtained. The line regulation of the proposed circuit is 0.434%/V with power consumption of 54.12 µW at room temperature.

## 1. Introduction

AIoT microchips are designed to enable Artificial Intelligence (AI) and Internet of Things (IoT) capabilities in a single device. AIoT chips are typically designed to handle the computational demands of AI algorithms while providing interfaces and protocols to connect with IoT devices and networks. The integration of complex neural networks to perform complex AI computations efficiently demands a precise and low-power reference voltage generator on the chip.

CMOS and bandgap reference are the two types of circuits widely used to provide a stable reference voltage through the process, voltage, and temperature (PVT) changes. MOSFET devices will be used in the CMOS reference circuit to represent the complementary to absolute temperature (CTAT) [1,2] and proportional to absolute temperature (PTAT) [3] characteristics. The MOS transistor’s threshold voltage, Vth, determines the reference voltage while holding the CTAT characteristic with minimal temperature dependence on BJT. The simplified CMOS-based conventional voltage reference generator is depicted in Figure 1.

The reference voltage is the voltage difference between the gate sources of two MOS transistors [4]. Both transistors are operating in the saturation region. Equation (Equation 1) expresses the *V*_REF_ from the two MOS transistors.
(1)$$V_{\mathrm{REF}}=V_{\mathrm{GS}2}-V_{\mathrm{GS}1}=V_{\mathrm{th}2}-V_{\mathrm{th}1}-\sqrt{2I}\left(\sqrt{\frac{1}{K_{2}}}-\sqrt{\frac{1}{K_{1}}}\right)$$

In CMOS reference, the dependency of reference voltage on temperature can be reduced by lowering the bias current, *I* [5]. In this condition, the reference voltage is approximately *V*_REF_ = *V*_th2_ − *V*_th1_. Usage of MOSFET in the reference design of CMOS will be an added advantage, as it consumes less power and allows a reduction in chip area. However, MOSFET is less sensitive to temperature, and thus it requires multiple trimming points across the process. The bandgap reference circuit has been adjusted during fabrication to produce the desired output voltage at a specific temperature. Multiple-point trimming calibrates the circuit at numerous temperature points to increase accuracy and thermal stability across a wider temperature range.

In the BGR circuit, the BJT has been used as a diode where the p-n junction of the diode is coupled with an intrinsic silicon bandgap voltage, *V*_BG_. When the bias current is applied to the p-n junction, it produces CTAT voltage with a substantial temperature dependency. On the other hand, the design parameters can be used to adjust the temperature coefficient of a PTAT voltage to generate VBG as well [6]. Since the BJT has higher temperature sensitivity and higher negative temperature co-efficient, it is good to inherit the replication of CTAT behavior in BGR despite it occupying a larger area on the chip and draining higher current [5]. Besides that, the BJT transistor is the best to compensate for process variation, as it holds a single-point trimming compared to the MOSFET transistor.

As compared to the BGR architecture, the CMOS reference voltage generator has lower power consumption and occupies a smaller area. However, due to the additional *V*_th_ as it outputs voltage headroom, it is subject to process variation. Various solutions have been proposed to solve this issue without trimming circuits, such as hybrid architecture, geometry dependence [7], and process compensation scheme [8,9]. However, all these solutions have been traded off to MOSFET’s temperature coefficient.

Various techniques have been proposed to improvise the performances of BGR in the aspect of the area, power consumption, and trimming. In [10], bandgap voltage and current references (BGVCR) technique, without an amplifier, was proposed to reduce the chip area. To produce PTAT current to guarantee the stability of the system, an amplifier was used in most BGR designs [11]. The amplifier takes up space on the chip and degrades the accuracy of the reference voltage, owing to input offset voltage and noise. Despite eliminating the amplifier from the design, a relatively bigger area was still consumed on the chip. Apart from this, BGR with a PTAT-embedded amplifier was introduced to reduce the chip area [12]. It consists of a single current branch that draws lower power and only consumes an area of 0.0082 mm^2^. This method contributes to higher noise, as the load on the amplifier has been increased.

The resistor-less BGR circuit is another method that is widely used in BGR design. The resistor that was previously used to integrate the PTAT and CTAT voltage characteristics to create the reference voltage has been removed in this architecture. To compensate for the use of the resistor in BGR, single-branch floating PTAT voltage and PTAT voltage generators with voltage duplicator techniques were created in [13,14], respectively. The voltage of the bipolar transistor, *V*_EB_, was directly floating on CTAT and was biased by a resistor-less current source in single-branch PTAT voltage, cascoded high-impedance current bias techniques. In single-branch PTAT voltage, where the voltage of a bipolar transistor, *V*_EB_, was directly floated on CTAT and biased by a resistor-less current source, the cascoded high-impedance current bias techniques were utilized.

In [13], the proposed voltage duplicator was multiplied by four times of PTAT, which was produced by connecting two PMOS differential pairs in series. The input of the voltage duplicator receives the bipolar transistor’s CTAT voltage or *V*_EB_. This method aids in obviating the necessity for a resistor. The proposed methods, however, worsen the temperature coefficient performance since there is no suitable PTAT and CTAT combination to generate the reference voltage.

A resistor-less CMOS reference design has been developed to create lower power, high PSRR reference voltage for SoC applications in the MHz frequency ranges. Due to the lack of a resistor in the design, most architectures, unfortunately, have trouble providing higher-order compensation as temperature sweeps from low to high. A poor temperature coefficient results from this scenario. To address this problem, resistor-less BGR successive voltage step compensation was put forth in [15,16]. The PSRR is only guaranteed by this approach, however, for lower frequencies and temperature coefficients.

To overcome the aforementioned problems, this research suggests an auto-trimming BJT-based BGR with a PSRR improvement circuit. System stability is maintained without compromising BGR’s performance in terms of temperature coefficient, line regulation, and chip area. To solve the issue of process variance, the automatic trimming circuit with a straightforward comparator and network resistor has been added to the PTAT resistor. The comparator will detect when the reference voltage is out of alignment due to process variation because its input is connected to the reference voltage, and its output will activate the trimming network resistor based on the appropriate weighting to compensate for the desired output. The paper is organized as follows. The suggested BGR and circuit implementation for each block are described in Section 2. Meanwhile, Section 3 discusses the measurement data and analysis, and Section 4 provides a conclusion.

## 2. Proposed Bandgap Voltage Reference

Figure 2 depicts the ATBGR schematic. A BGR Core (CTAT, PTAT, and op-amp), a startup circuit, a PSRR enhancement circuit, and an auto-trimming circuit are integrated. The auto-trimming circuit generates a consistent reference voltage across CMOS process variations.

### 2.1. Core Circuit of Bandgap Voltage Reference

Figure 3 illustrates the proposed BGR core circuit.

To design BGR with a less sensitive solution across the PVT, the circuit needs to bias itself, and the current need to independently flow to all of three branches in Figure 3. The current mirror approach has been used to supply equal current across the CTAT, PTAT, and *V*_REF_ generator circuit. The relationship of CTAT and PTAT to *V*_REF_ is illustrated in Figure 4.

Referring to Figure 4, the correct pairing of the CTAT and PTAT can eliminate temperature-dependent variations and result in a stable and temperature-independent reference voltage. CTAT and PTAT node’s potential voltage is ideally equal, as shown in Equation (Equation 2).
(2)$$V_{D}=V_{2}$$

Since a bipolar junction transistor is employed in this design as a diode, the voltage across the diode is defined below in terms of the thermal voltage:(3)$$V_{D}=V_{T}ln\left(I_{O}/I_{S}\right)$$
where *I*_O_ and *I*_S_ are the BJT’s collector current and reverse saturated current. The CTAT voltage is designated *V*_D_, exhibiting a negative temperature coefficient (TC).

On the PTAT side, number, *n*, of the bipolar transistor was added to minimize the potential voltage difference between the CTAT and PTAT. The voltage across the diode is given in (4):(4)$$V_{D}=V_{T}ln\left(I_{O}/nI_{S}\right)$$

Using (3) and (4), the voltage across the resistor, *R*_1_, is derived as follows:(5)$$V_{R_{1}}=V_{D}-V_{D1}=V_{T}ln\left(I_{O}/I_{S}\right)-V_{T}ln\left(I_{O}/nI_{S}\right)=V_{T}ln\left(n\right)$$
where *V*_T_ = kT/q, k is Boltzman’s constant, and q is the charge of the electron. Referring to (5), the voltage across the resistor, *R*_1_, is the PTAT voltage, VPTAT. There are three branches in the core circuit, including PTAT, CTAT, and REF_GEN. All these three branches have equal bias currents, and the total current of this core circuit can be expressed in (6).
(6)$$I_{Total}=I_{CTAT}+I_{PTAT}+I_{REF\_GEN}$$

Hence, *V*_PTAT_ is
(7)$$V_{R_{1}}=I_{PTAT}R_{1}$$

Since the voltage across the resistor, *R*_1_, is computed using Equation (Equation 5), and the biased current will be determined based on the design specification, the resistor, *R*_1_, value can be obtained with Equation (Equation 8). Equation (Equation 8) is expressed by Substituting (7) into the result of (5).
(8)$$R_{1}=\frac{V_{T}ln\left(n\right)}{I_{PTAT}}$$

To achieve minimal voltage variation, a total of eight bipolar transistor-based diodes are used in this design.

By assuming *V*_CTAT_ = *V*_D_ = 0.7 V as the typical diode voltage, the reference voltage of BGR is the total voltage across *R*_2_ and *Q*_10_, where they represent the behavior of PTAT and CTAT, respectively.
(9)$$V_{REF}=V_{R_{2}}+V_{Q_{10}}$$

The reference voltage is derived in terms of thermal voltage, *V*_T_, and diode voltage, *V*_D_, as in Equation (Equation 10).
(10)$$V_{REF}=\alpha \left(V_{T}\right)+\beta \left(V_{D}\right)$$
where α and β are the weightage of PTAT and CTAT, respectively. Exhibiting zero temperature coefficient can be achieved by adding two items with the opposite temperature coefficients with the appropriate weight, as expressed in Equation (Equation 11).
(11)$$\frac{\delta V_{REF}}{\delta T}=\alpha \left(\frac{\delta V_{T}}{\delta T}\right)+\beta \left(\frac{\delta V_{D}}{\delta T}\right)=0$$

Since CTAT has only one diode, the CTAT weightage, β = 1. Based on [16], the (δ*V*_T_)/(δ*T*) = 85 µV/°C) and (δ*V*_D_)/(δ*T*) = −1.63 mV/°C), by substituting into (11), the PTAT weightage, α, will be computed as 18.82. The PTAT weightage, α, is equivalent to *V*_*R*_2__ because it replicates the behavior of the PTAT as it is defined in (9). The voltage across the resistor, *V*_*R*_2__, is expressed as in (12).
(12)$$V_{R_{2}}=I_{REF\_GEN}R_{2}$$

Since all three branches flow the same current, by substituting Equation (Equation 7) into Equation (Equation 12), *R*_2_ can be determined. *R*_1_ and *R*_2_ play a critical role in determining the reference value. A trimming circuit has been implemented on *R*_2_ to ensure the reference voltage is independent of the process variation.

### 2.2. Two-Stage Operational Amplifier with Active Miller Compensation (AMC)

In our BGR, a two-stage op-amp is used to achieve the design goal of equal CTAT and PTAT potential differences. The schematic of the design is illustrated in Figure 5. If the amplifier detects inequality, the output of the amplifier will trigger the gate of current mirror M_8_ and M_9_ to increase the drain current to equalize the voltage differences. Figure 6 illustrates the simulated input voltages of the amplifier across the temperature. The simulated input voltages of the amplifier exhibit linear characteristics from −40 °C to 125 °C while generating the reference voltage with CTAT and PTAT. A stable and constant output voltage is produced because CTAT and PTAT are tied to each amplifier node.

The two-stage op-amp consists of a differential single-ended output with current mirror biasing as the first stage and a common source stage as the second stage. Its transfer function is given as follows:(13)$$H\left(s\right)=\frac{K(1+\frac{s}{\omega_{Z}})}{\left(1+\frac{s}{\omega_{P1}}\right)\left(1+\frac{s}{\omega_{P2}}\right)}$$

The gain of the op-amp is expressed as follows: (14)$$DCGain=gm_{M_{1}}*gm_{M_{10}}*(ro_{M_{2}}//\left(ro_{M_{9}}\right)*(ro_{M_{5}}//\left(ro_{M_{10}}\right)$$

Referring to Figure 5, the integrated M_F_ acts as an AMC for the high-gain op-amp. Operating comfortably in the saturation region due to high overdrive gate voltage from VDD contributes to higher active RC, which is inherited from M_F_. Higher RC moves the dominant pole away towards low frequency, thus improving the phase margin. Figure 7 illustrates the simulation results of the two-stage op-amp’s corresponding open-loop gain and phase margin. The maximum achieved is 64 dB with a phase margin of 60°.

The active miller compensation with integrated M_F_ helps to move the dominant pole to a higher frequency, that is, from 40 MHz to 50 MHz, and achieves a phase margin of 60 degrees, which is a stable condition.

### 2.3. Startup Circuit

Figure 8 illustrates the proposed startup circuit for the ATBGR.

The startup circuit consists of three PMOS transistors, a NMOS transistor, and a resistor, *R*_3_, designed to break the zero-current region into the normal operating region. *V*_N_ of M_1_ is connected to the startup circuit.

When the BGR core circuit is turned to normal operation mode, all transistors in the startup circuit are in hibernate mode, as condition *V*_N_ is zero current state. Only the transistor, M_18_, is ON, and the current starts to flow on the *V*_N_ node.

As a result, the amplifier will perform a comparison between *V*_N_ and *V*_P_ nodes, respectively, and correct another node by increasing the current flow through M_12_ of the core circuit, leading to an equal flow of current on both of nodes. As the input node of the amplifier starts to flow the current, the gate voltage of the M_19_ is in a “HIGH” state, enabling current mirror pair M_16_–M_17_. As a result, M_18_ is turned off, thus switching the startup circuit to hibernating mode. Figure 9 illustrates the simulated transient response of the ATBGR with and without the startup circuit.

The ATBGR quickly reaches a steady state after the startup circuit exits the zero-current region. On the other hand, without the startup circuit, it takes longer to reach a steady state, as depicted in Figure 9, as the amplifier’s input node was initiated by itself to break the zero-current region.

### 2.4. Auto-Trimming Circuit

Figure 10 illustrates the auto-trimming circuit integrated into the ATBGR to resolve the reference voltage variation issue across the process. *R*_2_ is used for the trimming purpose, referring to (12). A 4-bit trimming circuit has been integrated with *R*_2_. The value of the resistor is an increment from one to another in multiples of two, that is, *R*, 2*R*, 4*R*, and 8*R*, each connected in parallel to the PMOS switches M_20_–M_23_. PMOS is favorable over NMOS, as it exhibits reduced process sensitivity.

The trimming action is automated in the auto-trimming circuit (ATC) through built-in op-amps 1 and 2. The op-amps compare an external voltage, +1.25V_EXT, 1.25 V, with the reference voltage, *V*_REF_, to detect and reduce the variation. This external voltage was supplied from an external DC power supply model (Agilent-E3631A). In this design, two op-amps were used to capture the variation range.

The first op-amp was used to record any events where the output voltage was marginally higher than the external voltage. When the reference voltage is greater than the external voltage, the op-amp will be set. The negative op-amp node is connected and attached to the external voltage, +1.25V_EXT, while a positive node is connected to the *V*_REF_ output of the BGR core.

Op-amp 2 is used to determine whether the output voltage was less than the external voltage. When the output voltage is less than the external voltage, the op-amp will set. The output of the BGR was connected to its negative node and an external voltage, +1.25V_EXT, is connected to the positive node. When the output voltage of the BGR exceeds the external reference voltage, the resistors R and 2R are configured, whereas larger resistances, such as 4R and 8R, are configured when the output voltage is lower. The transistor-level schematic of the op-amp is shown in Figure 11.

The 2-to-4 demultiplexer is used to control the resistor network through the PMOS switches. The output of the op-amps is used as the selection pin of the de-multiplexer. The MOSFET switches will be configured according to the response of the op-amp. The trimming step for the 4-bit trimming circuit is 110 µV/LSB. The truth table of the 2-to-4 demultiplexer is shown in Table 1.

To verify the effectiveness of the ATC, a Monte Carlo simulation has been performed for 200 samples. The Monte Carlo simulation was used to analyze the normal distribution of both with and without the ATC, and the results are illustrated in Figure 12 and Figure 13, respectively.

Referring to Figure 12 and Figure 13, the BGR with ATC achieves *V*_REF_ with higher precision (1.25001 V) than without ATC (1.25156 V). According to the findings, the ATBGR with ATC-distributed data is close to the mean value compared to the ATBGR without ATC. ATC has resulted in a normal distribution. Furthermore, the latter has a lower standard deviation than the former.

### 2.5. PSRR Enhancement Circuit

To enhance the PSRR performances in the BGR circuit at a higher frequency range, a cascaded PMOS common gate pair has been integrated at the output of the BGR, as illustrated in Figure 2.

An analysis model of the BGR is illustrated in the Figure 14 below.

In the figure, *Z*_out_ is the output impedance of the BGR and *Z*_BGR_ is the shunting effect of its feedback loop. *Z*_out_ and *Z*_BGR_ are given as follows: (15)$$Z_{out}=Z_{cl}\left|\right|R_{out}$$
(16)$$R_{out}=r_{ds14}+r_{ds15}+R_{1}\left|\right|R_{2}+r_{ce10}$$
(17)$$Z_{BGR}=\frac{Z_{out}\left|\right|r_{ds}^{\prime }}{{}^{*}A_{o1}.\frac{r_{ce10}}{(R_{1}\left|\right|R_{2})+r_{ce10}}}$$
where
(18)$${}^{*}A_{o1}=g_{m14}r_{o14}.g_{m15}r_{o15}$$
and
(19)$$r_{ds}^{\prime }=r_{ds11}\left|\right|r_{ds12}\left|\right|r_{ds13}$$

From here, *V*_REF_ is calculated as
(20)$$V_{REF}=\frac{Z_{out}\left|\right|Z_{BGR}}{r_{ds}^{\prime }+Z_{out}\left|\right|Z_{BGR}}.V_{DD}$$
and PSRR is obtained as
(21)$$PSRR=\frac{V_{REF}}{V_{DD}}=\frac{Z_{out}\left|\right|Z_{BGR}}{r_{ds}^{\prime }+Z_{out}\left|\right|Z_{BGR}}$$

The final equation is given in the Appendix A.

The simulation results of the PSRR with and without the PSRR enhancement circuit are illustrated in Figure 15. It can be observed that the enhancement circuit improves the rejection significantly at a frequency of more than 1 KHz.

## 3. Measurement Results

The proposed BJT-based BGR has been designed and fabricated in 0.18 µm CMOS technology. The chip area of the BGR is 0.032768 mm^2^, including the bond pads for measurement. Figure 16 depicts the photomicrograph of the proposed BJT-based BGR with the bond pads.

To validate the proposed BGR design, a total of 10 samples of the chip were measured. The 10 samples are selected from various wafers that fall under the FF, TT, and SS speed grades. Figure 17 shows the output voltage of BGR with a temperature range from −40 °C to 125 °C with a supply voltage of 3.3 V. The BGR was designed to generate 1.25 V as an output voltage. The output voltage of the proposed BGR has a deviation smaller than 1.278 mV across the temperature ranging from −40 °C to 125 °C. Based on the 10 samples, the minimum and maximum temperature coefficients are 6.11 ppm/°C and 15.32 ppm/°C, respectively, with power supply ranges from 1.65 V to 4.5 V. The proposed BGR core circuit only consumes 16.4 µA when the input supply is 3.3 V.

Additionally, the input voltage variation is also taken into consideration in this proposed BGR circuit. Figure 18 shows the output voltage of BGR with swiping the input supply from 0 V to 4.5 V. The output voltage starts to become saturated when the input supply is 1.65 V and remains in the saturation region up to 4.5 V. During this saturation period, the deviation of the output voltage is only 15.47 mV. The measured line regulation of the output voltage is 0.434%/V across the power supply of 1.65 V to 4.5 V.

The performance distribution of the proposed BGR for the ten samples after trimming was extracted from Figure 17 and Figure 18, and the distribution of *V*_REF_ and the temperature coefficient is shown in Figure 19 and Figure 20, respectively. Figure 19 illustrates the measured mean output reference voltage at room temperature, that is, 1.2495 V with a standard deviation of 2.38 mV. The measured TC distribution is depicted in Figure 20 and has a mean value of 9.99 ppm/°C and a standard deviation of 3.06 ppm/°C.

The measured results of the PSSR of the proposed BGR circuit are shown in Figure 21. The PMOS PSRR enhancement circuit helps to optimize the PSRR of the BGR circuit across the frequency, especially in the higher frequency range.

The measurement results are compared against other BGRs from the literature in Table 2. With a supply headroom range of 1.65 V to 4.5 V and a reasonable temperature coefficient, the proposed BGR provides the best PSRR at higher frequency ranges.

## 4. Conclusions

An auto-trimming BJT-based bandgap voltage reference, ATBGR, with a PSRR improvement circuit is proposed in this work. Even though the auto-trimming and PSRR improvement circuits were included in this design, the proposed BGR takes up less space. By optimizing a two-stage differential amplifier and including an NMOS transistor in the second output stage, the stability of the system has been improved. A competitive advantage has been achieved, thus qualifying the ATBGR in application designs such as System on Chip (SoC), mobile devices, medical implants, Internet of Things (IoT), and Wireless Sensor Node (WSN) due to the high supply rejection ratio that was accomplished at a higher frequency range by the PSRR augmentation circuit. The proposed circuit has been fabricated using 180 nm CMOS technology with an area of 0.327768 mm^2^. The average reference voltage and temperature coefficient across temperature ranges from −40 °C to 125 °C are 1.25 V and 6.49 ppm/°C, respectively, when the supply voltage is within the range of 1.65 V to 4.5 V. Line regulation of the proposed BGR is 0.424%/V across the supply voltage range and the PSRR is −90 dB at 100 MHz.

## Figures and Tables

**Figure 1 micromachines-14-01724-f001:**
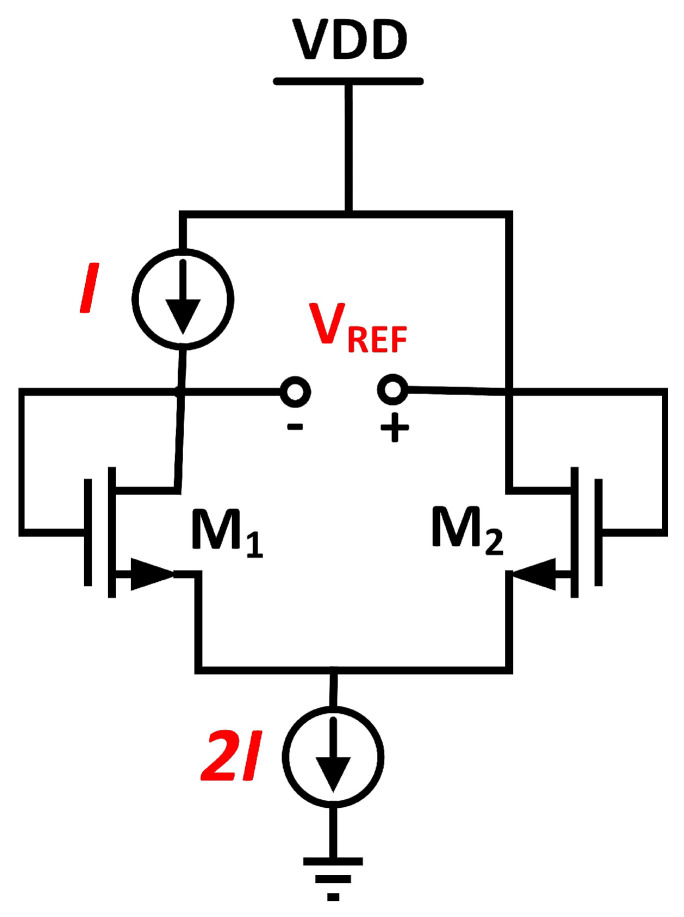
Simplified CMOS−based voltage reference generator.

**Figure 2 micromachines-14-01724-f002:**
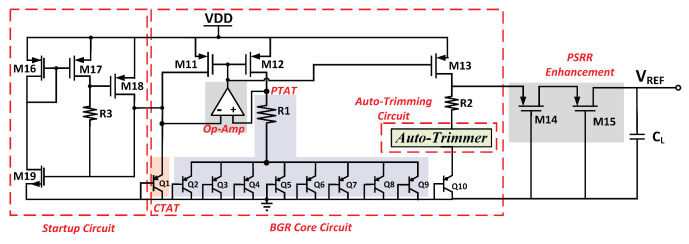
Schematic of proposed BGR circuit.

**Figure 3 micromachines-14-01724-f003:**
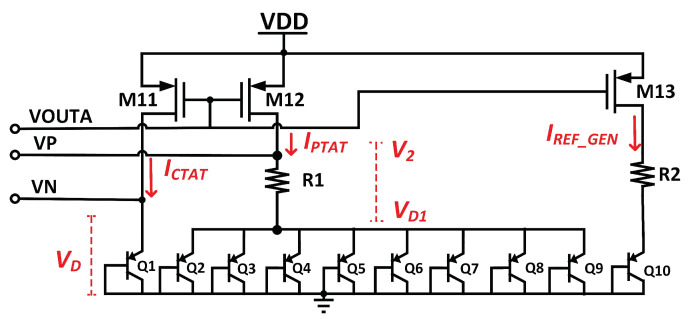
Schematic of proposed BGR core circuit.

**Figure 4 micromachines-14-01724-f004:**
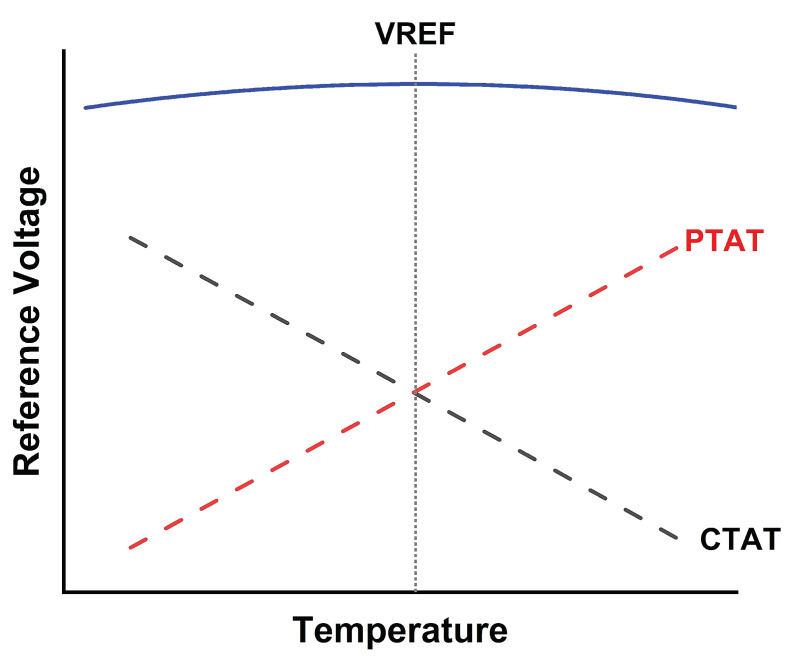
Relation of CTAT and PTAT.

**Figure 5 micromachines-14-01724-f005:**
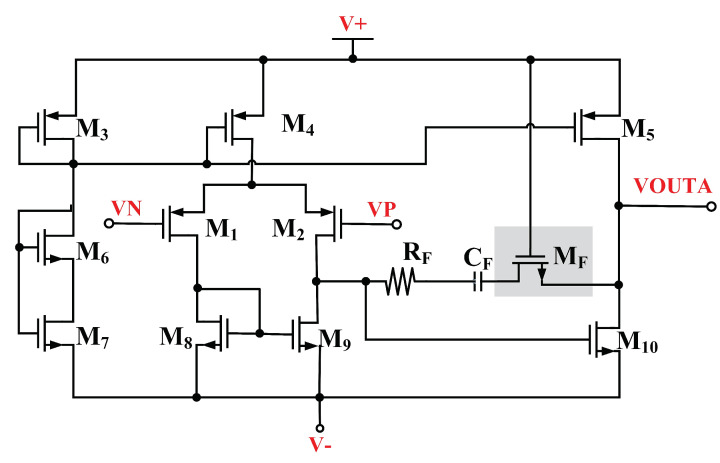
Schematic of the two-stage differential amplifier.

**Figure 6 micromachines-14-01724-f006:**
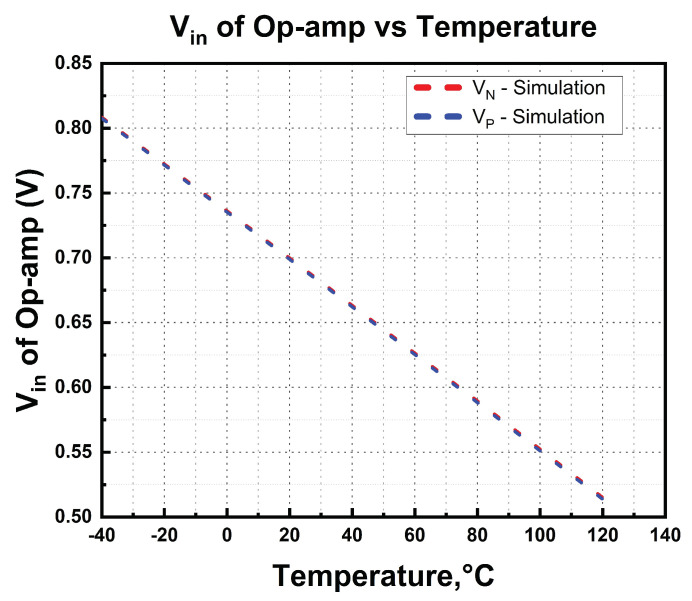
Simulated input voltage of the two-stage differential amplifier.

**Figure 7 micromachines-14-01724-f007:**
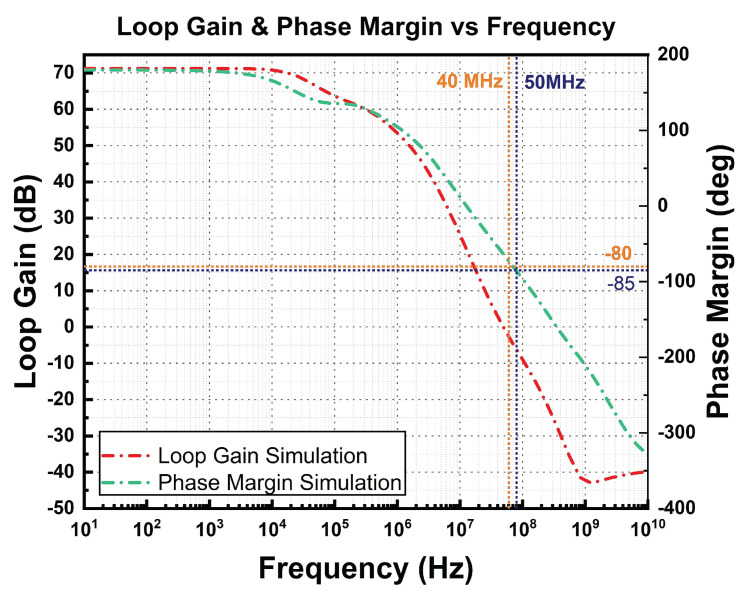
Simulated results of open−loop gain and phase margin of two-stage op-amp.

**Figure 8 micromachines-14-01724-f008:**
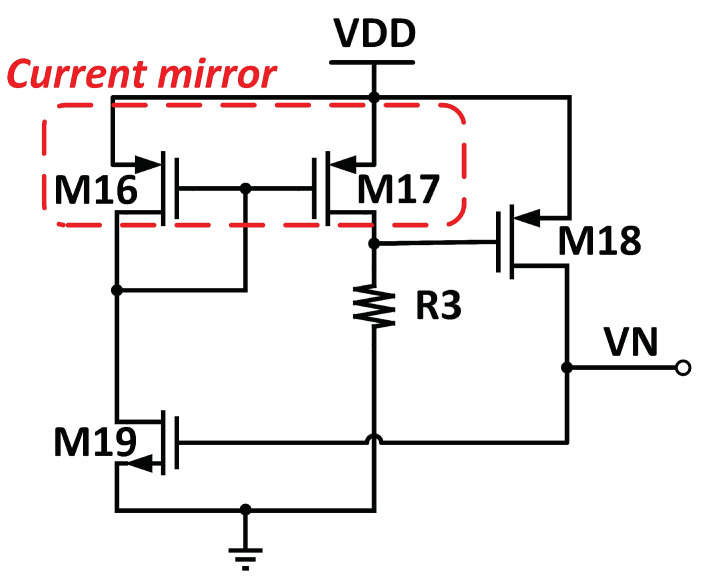
Startup circuit of the ATBGR.

**Figure 9 micromachines-14-01724-f009:**
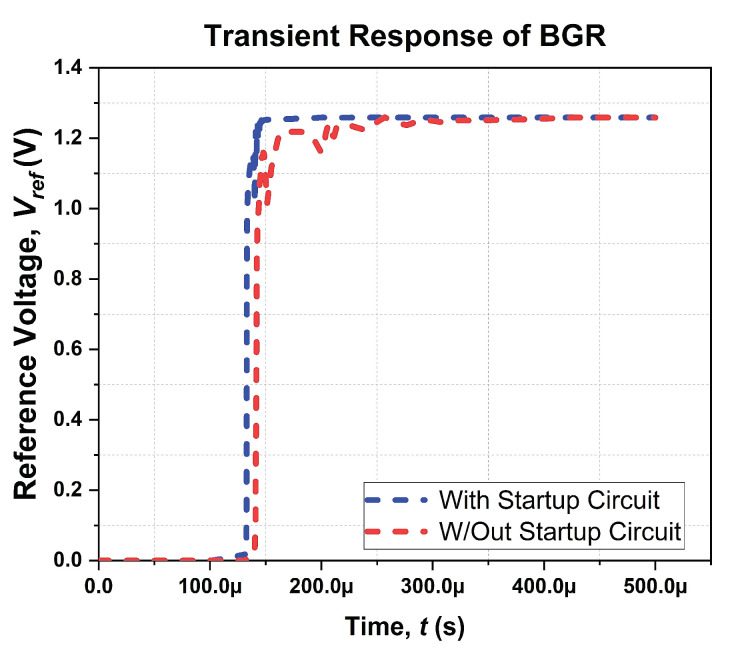
Simulated transient response of BGR with and without startup circuit.

**Figure 10 micromachines-14-01724-f010:**
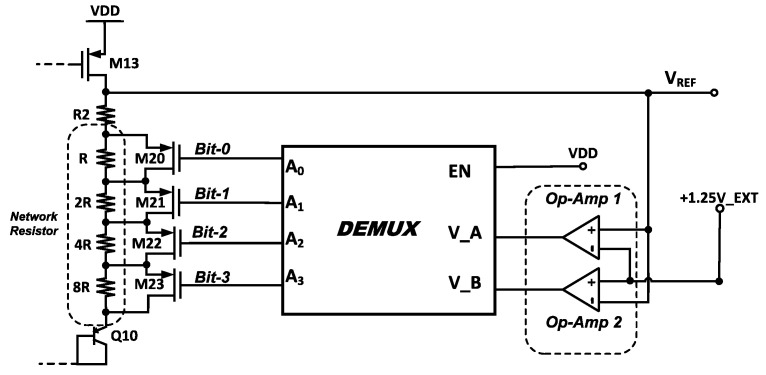
Schematic of auto-trimming circuit.

**Figure 11 micromachines-14-01724-f011:**
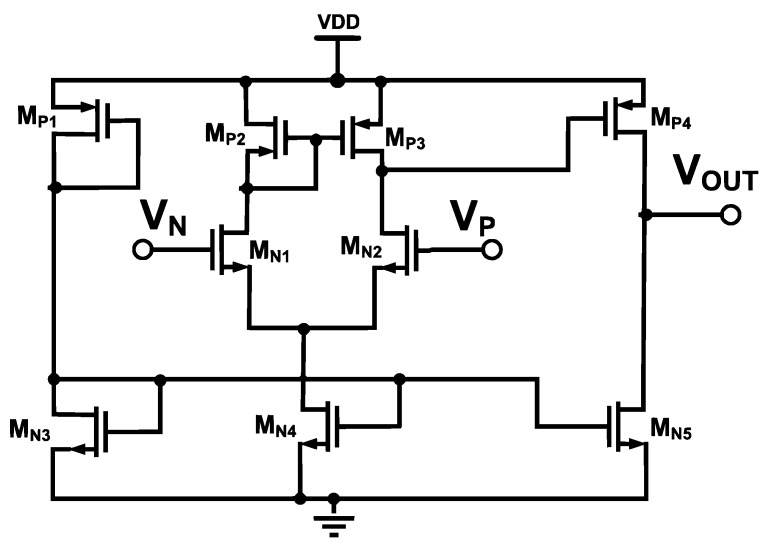
Schematic of operational amplifier.

**Figure 12 micromachines-14-01724-f012:**
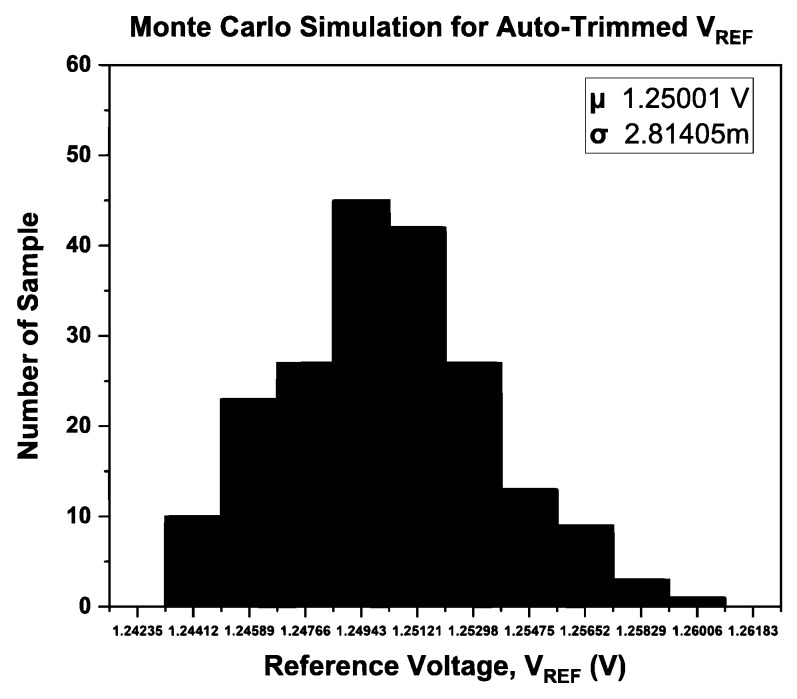
Monte Carlo simulation results for BGR with ATC.

**Figure 13 micromachines-14-01724-f013:**
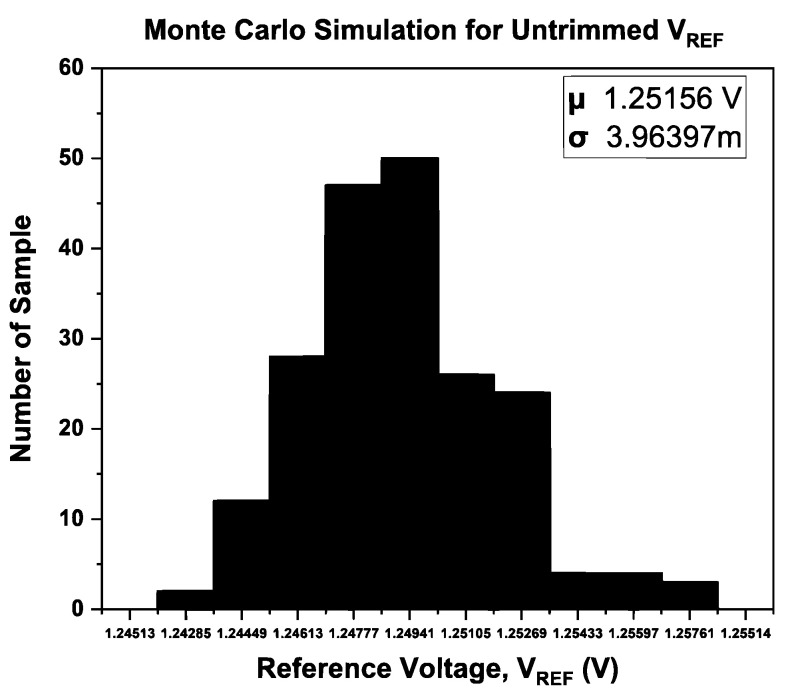
Monte Carlo simulation results for BGR without ATC.

**Figure 14 micromachines-14-01724-f014:**
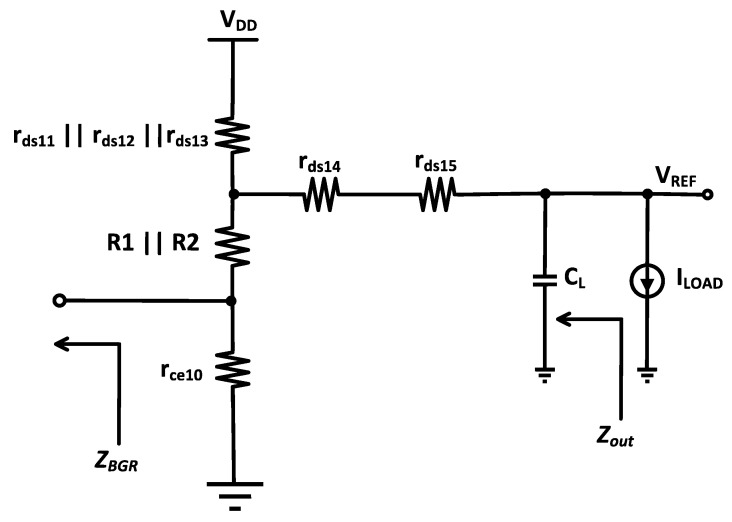
Model of proposed BGR.

**Figure 15 micromachines-14-01724-f015:**
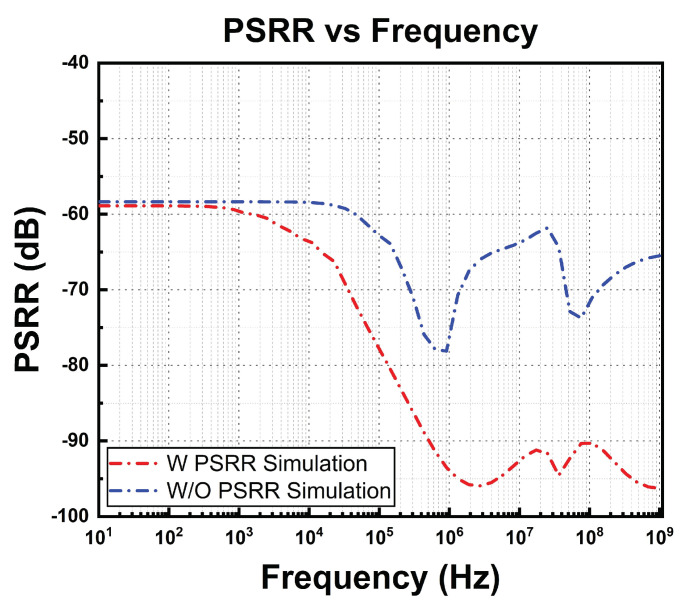
Simulated PSRR of BGR with and without PSRR enhancement circuit.

**Figure 16 micromachines-14-01724-f016:**
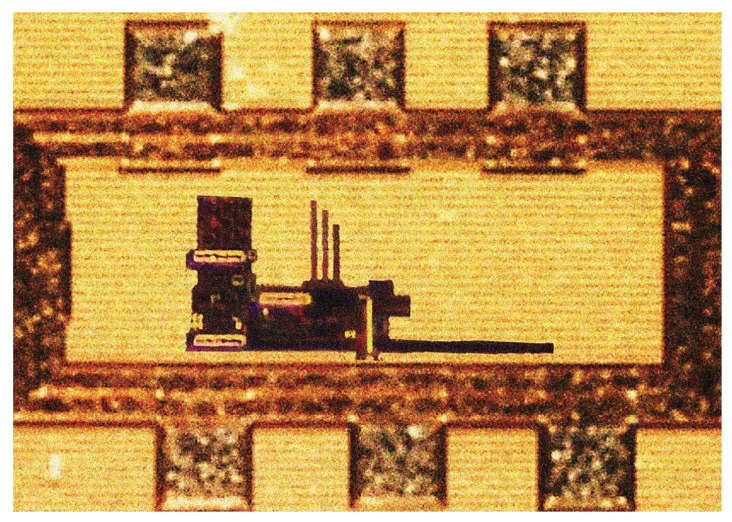
Micrograph of the designed BJT−based BGR.

**Figure 17 micromachines-14-01724-f017:**
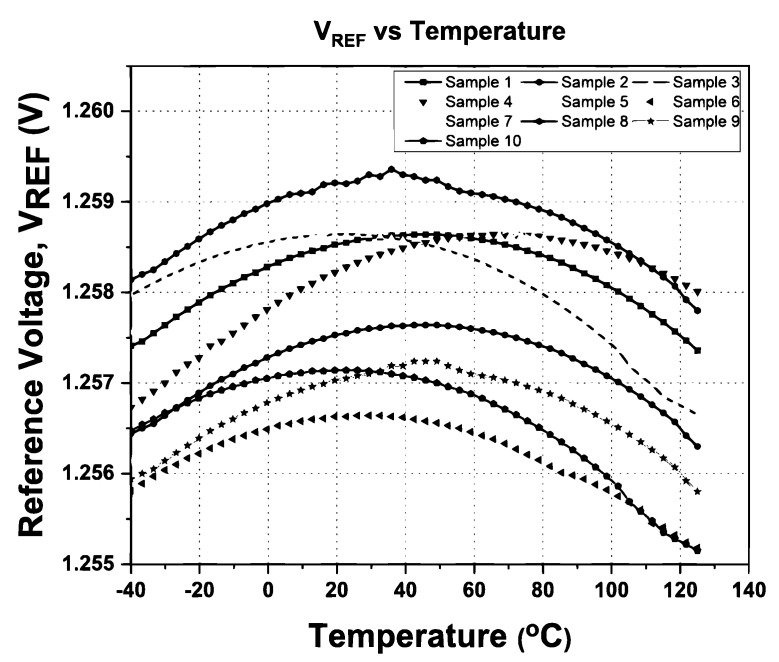
The output voltage of BGR across temperature.

**Figure 18 micromachines-14-01724-f018:**
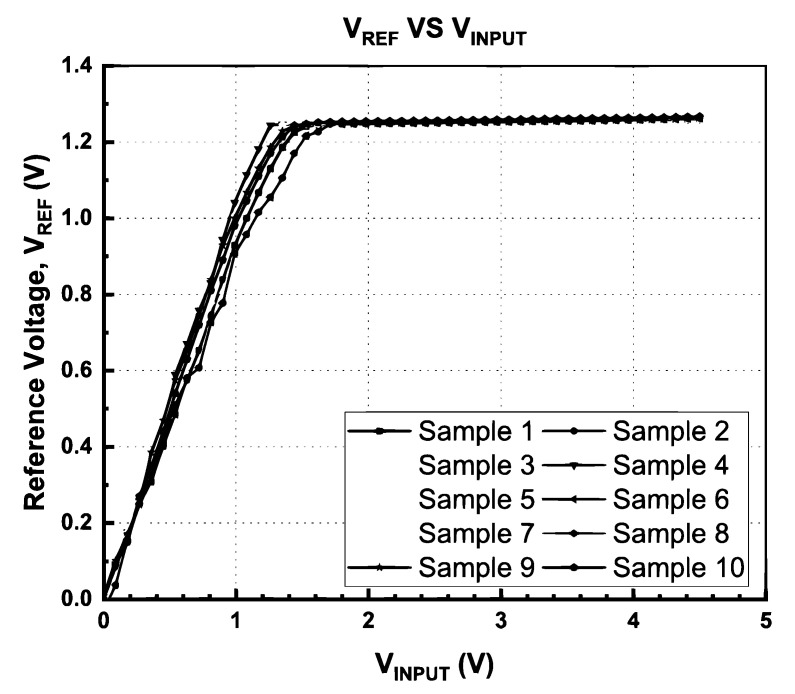
Theoutput voltage of BGR across input voltage.

**Figure 19 micromachines-14-01724-f019:**
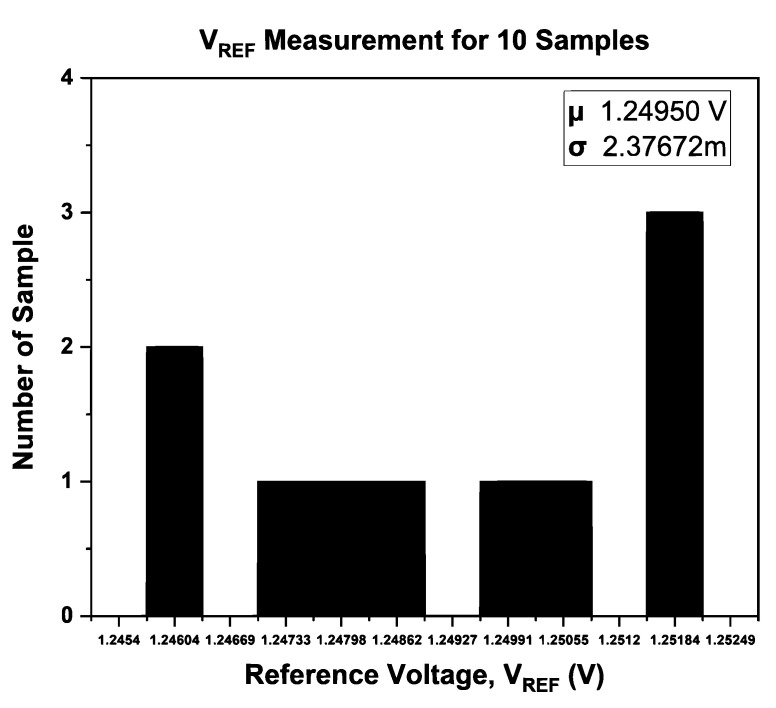
Distribution of *V*_REF_ for 10 samples at room temperature.

**Figure 20 micromachines-14-01724-f020:**
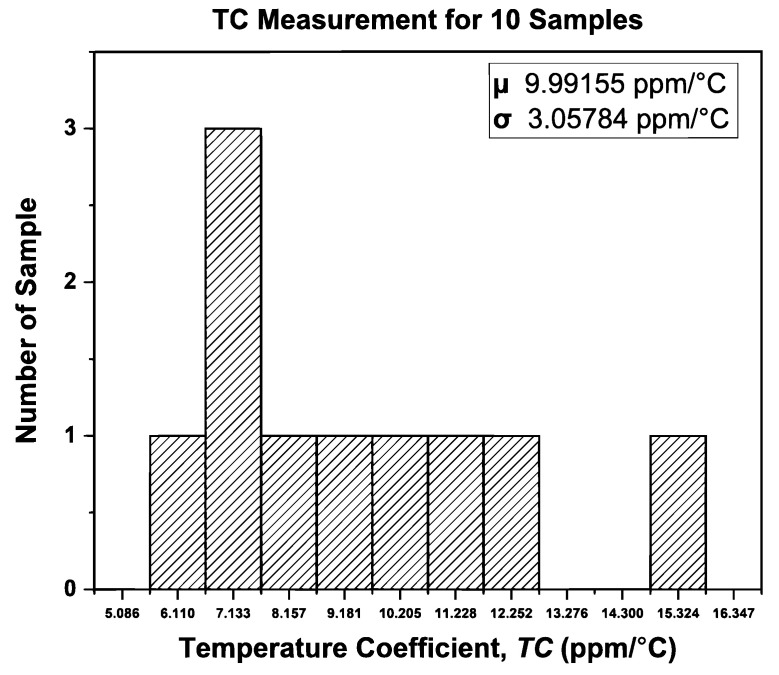
Distribution of TC for 10 samples.

**Figure 21 micromachines-14-01724-f021:**
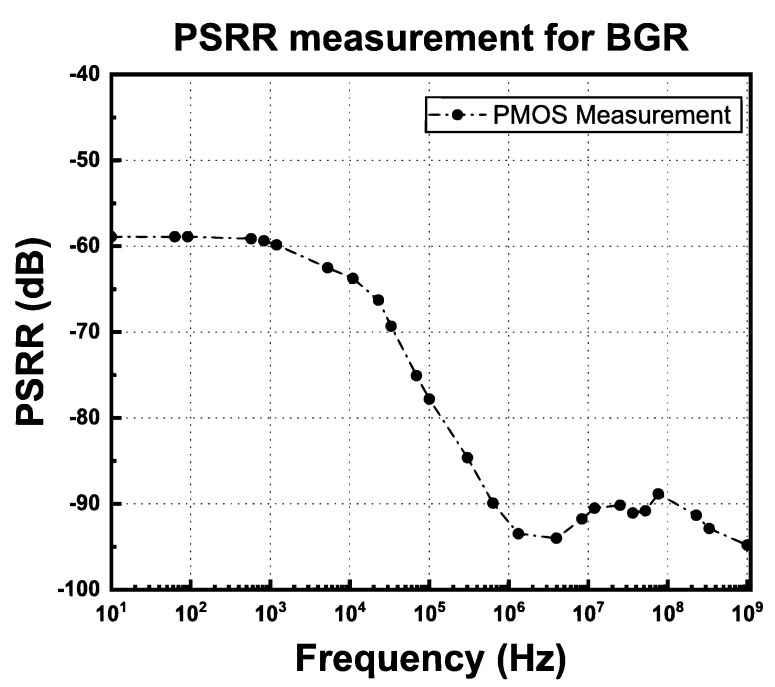
Measured PSRR of BGR with PMOS PSRR enhancement circuit.

**Table 1 micromachines-14-01724-t001:** Truth table of demultiplexer of auto-trimming circuit.

EN	V_A	V_B	A_0_	A_1_	A_2_	A_3_	Resistance
0	*X*	*X*	0	0	0	0	*R* _2_
1	0	0	1	0	0	0	*R*_2_ + *R*
1	0	1	0	1	0	0	*R*_2_ + 2*R*
1	1	0	0	0	1	0	*R*_2_ + 4*R*
1	1	1	0	0	0	1	*R*_2_ + 8*R*

**Table 2 micromachines-14-01724-t002:** Summary of the performance of the proposed ATBGR compared with other state-of-the-art BGRs.

Reference	This Work	[4]	[13]	[14]	[15]	[17]	[18]
Year	2023	2019	2019	2020	2020	2022	2022
Technology (nm)	180	180	350	180	130	350	65
Temperature Range (°C)	−40 to 125	−20 to 100	−40 to 125	−40 to 140	−20 to 110	0 to 80	−40 to 120
Supply Voltage, *V*_IN_ (V)	1.65–4.5	1.0	2.0–5.0	1.3–1.8	1.4	2.8–4.5	0.5
Supply Current, *I*_IN_ (A)	16.4µ	0.192n	33µ	107n	5.9n	18n	76n
Power Consumption, P (W)	27.1µ	192p	66µ	139.1n	8.26n	50.4n	38n
Reference Voltage, *V*_REF_ (V)	1.25	0.6926	1.14055	1.17	1.252	1.17	0.495
TC (ppm/°C)	6.49	33	1.01	26.3	68	65	42
PSRR (dB) @ 100 Hz	−58	−55	−61	−52	-	-	−50 @ DC
PSRR (dB) @ 1 MHz	−93	-	-	−44	-	-	-
PSRR (dB) @ 100 MHz	−58	-	-	-	-	-	-
Noise @ 10 Hz (µV/sqr (Hz)) (dB) @ 100 MHz	34.6	26.8	-	-	6	-	-
Line Regulation (%/V)	0.424	0.02	2 mV/V	0.07	0.019	0.112	0.64
Chip Area (mm^2^)	0.033	0.0045	0.0396	0.0082	0.017	0.042	0.0532

## Data Availability

Not applicable.

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
