# Peer review of "A 54 µW CMOS Auto-Trimming Bandgap References (ATBGR) Achieving 90 dB PSRR for Artificial Intelligence of Things (AIoT) Chips"

_micromachines, 2023, doi:10.3390/mi14091724_

Round 1

Reviewer 1 Report

It is suggested to explain more about the proposed technique in the Abstract Section.

The number of keywords seems more than usual.

How are the dimensions of the devices used in the presented circuits chosen?

For some of the performed simulations, more explanation and interpretation is required.

Is it possible to provide a better and clearer picture of the fabricated chip?

Is there a possibility to define a FoM by the authors to better compare this work and similar works?

The paper shows the results of a Monte Carlo analysis. What individual distribution laws are defined in the Monte Carlo analysis?

Author Response

Hi Reviewer,

Thank you very much for all the constructive comments. We have addressed all of them and improved the quality of the manuscript. Please refer to the attachment for the point-by-point response. 

Thank you very much.

Reviewer 2 Report

Please explain further the meaning of sigle/multiple point trimming.

R2 is mentioned before its presentation in section 2 (refer to page 3).

Equation (5): correct ln in 2nd and third terms.

Fig. 6 is not mentioned in the text, please explain its meaning and purpose.

Fig. 7: Please confirm in text and in the figure caption that this is open-loop gain (is it?).

Please comment and give an extra insight on how the VDD at the left in figure 10 is somehow other than the VDD supply of the CMOS chip. With VDD variation, the input for opamp2 will not be exactly 1.25. How this variation affects the data shown in the proposal?

Please include power (not only current consumption) in the comparison table 2.

English must be improved; several typos and more than a few verbs and lack of adjectives in the document. The word Abstract is repeated. 

Authors must avoid ambiguos quantification, like: wise, good, bigger, confortably, etc ...

Author Response

(The authors gave the same response as above.)

Reviewer 3 Report

An auto trimming BJT-based bandgap voltage reference with PSRR improvement circuit is proposed in this work. Even though the auto trimming and PSRR improvement circuits were included in the this design, the proposed BGR takes up less space. By optimizing a two-stage differential amplifier and including an NMOS transistor in the second output stage, the stability of the system has been improved. Authors claim to achieve a competitive advantage to adopt their BGR in application designs such as System on Chip (SoC), mobile device, medical implant, Internet of Things (IoT), and Wireless Sensor Node (WSN) due to the high supply rejection ratio that was accomplished at a higher frequency range by PSRR augmentation circuit. 

The proposed circuit has been fabricated using a 180nm CMOS technology with an area of 0.327768mm2.

Measurements results are compared against other BGRs from the literature in Table 2.

The paper is enough interesting, clear and well written, even if the english form can be improved in many points.

Main concern of this reviewer is related to the autotrimming circuit. More specifically authors in Section 2.4 state to compare the Vref of the BGR with a +1.25V reference derived from a voltage divider (see Fig. 10). But in this way the +1.25 V is sensitive to the supply voltage VDD which could vary a lot during operation. Authors should add specific comments on this point.ù

Another minor remark is related to the MOS devices M20-M23 in Fig. 10. In the text it is said that they are PMOS transistors, but the symbol in Fig. 10 seems wrong.

The paper is enough interesting, clear and well written, even if the english form can be improved in many points. Please revise the english throughout the paper.

Author Response

(The authors gave the same response as above.)
